# The effect of information content on acceptance of cultured meat in a tasting context

**Nathalie C. M. Rolland[1,2], C. Rob Markus[3], Mark J. Post[1,4]***

**1** Department of Physiology, Maastricht University, Maastricht, The Netherlands, **2** Department of Nutrition and Food Science, AgriSup, Université Bourgogne Franche-Comté, Dijon, France, **3** Department of Psychology, Maastricht University, Maastricht, The Netherlands, **4** Mosa Meat, B.V., Maastricht, The Netherlands

\* m.post@maastrichtuniversity.nl

## Abstract

Cultured meat, in particular beef, is an emerging food technology potentially challenged by issues of consumer acceptance. To understand drivers of consumer acceptance as well as sensory perception of cultured meat, we investigated the effect of information content on participants' acceptance of cultured meat in a tasting context. Hundred ninety-three citizens from the Netherlands participated, divided across three age and sex-matched groups which each received information on either societal benefits, personal benefits or information on the quality and taste of cultured meat. They filled out a questionnaire and tasted two pieces of hamburger, labeled 'conventional' or 'cultured', although both pieces were in fact conventional. Sensory analysis of both hamburgers was performed. We observed that provision of information and the tasting experience increased acceptance of cultured meat and that information on personal benefits of cultured meat increased acceptance more than information on quality and taste but not than societal benefits of cultured meat. Previous awareness of cultured meat was the best predictor of its acceptance. In contrast to previous studies, sex and social economic status were not associated with different acceptance rates. Surprisingly, 58% of the respondents were willing to pay a premium for cultured meat of, on average, 37% above the price of regular meat. All participants tasted the 'cultured' hamburger and evaluated its taste to be better than the conventional one in spite of the absence of an objective difference. This is the first acceptance study of cultured meat where participants were offered to eat and evaluate meat that was labeled 'cultured'. We conclude that having positive information importantly improves acceptance and willingness to taste and that the specific content of the information is of subordinate importance. Awareness of cultured meat is the best predictor of acceptance.

## Introduction

Cultured meat aka "clean" or "cell-based" meat, is an emerging food technology that has potential benefits for resource efficiency, environmental impact and animal welfare compared

**Data Availability Statement:** Data are found in a Mendeley repository, URL: https://data.mendeley.com/datasets/658544trwd/1.

**Funding:** This study was funded by: MJP received an award from Stichting Wetenschappelijk Onderzoek Limburg. It has no grant number. URL:

https://www.ufl-swol.nl. Zandbergen World's Finest Meats (Zoeterwoude, The Netherlands) provided the hamburgers. The funder had no role in study design, data collection and analysis, decision to publish, or preparation of the manuscript.

**Competing interests:** The authors have read the journal's policy and the authors of this paper have the following competing interests: Mark J. Post is Chief Scientific Officer and co-founder of Mosa Meat, a company that aims to commercialise cultured meat. There are no patents, products in development or marketed products associated with this research to declare. This does not alter our adherence to PLOS ONE policies on sharing data and materials.

to livestock beef production [1, 2]. Cultured beef is meat grown in a lab from bovine muscle specific stem cells. Several companies are currently developing products based on this technology, which has also been extended to other species including pork, chicken and fish. During introduction of this technology to the public, it became clear that public acceptance was not immediate and perhaps not obvious. The nature of this hesitance is poorly understood, although qualitative group interviews and initial surveys have shed some light on the arguments and description of feelings that form the public discourse on cultured meat [3–6].

The theoretical framework on rejection of novel and unfamiliar foods was laid down by Rozin and Fallon [7], who suggested a taxonomy of food rejections based on taste or sensory perception in general, danger (safety, health) and disgust. Disgust is a mostly emotional response determined by cultural connotations or relatively fixed ideas of what is an 'appropriate' food or not. The disgust response is typically stronger for novel animal-based products than for non-animal products [7]. In a more recent study on acceptance of novel foods, Martins and Pliner [8] identified the disgust factor as being an important determinant in willingness to try these products.

Although the disgust response is an emotional one, it is based on culturally defined ideas and therefore considered 'cognitive'. This element of rejection or acceptance may thus be influenced by information provided about the novel food, particularly if the novel food is highly technological [9]. There are preliminary indications that providing positive information on cultured meat increases the self-reported willingness to try, buy or pay more for cultured meat [3]. In an early experiment, conducted in 2012, Bekker et al. [10] explored the effect of positive and negative information on implicit (e.g. disgust) and explicit attitudes (e.g. willingness to buy) towards cultured meat and found that this information indeed positively or negatively affected explicit attitudes towards cultured meat but had no effect on implicit attitudes. Both these studies were performed with university students in agricultural or food science and may not be generalizable to the population at large.

The present study aims to answer the question if specific information on societal or personal benefits or information about the quality of the food-product affects consumer acceptance and sensory perception of cultured meat in a general population. This is the first acceptance study that includes a tasting experiment with a piece of meat labeled as 'cultured'.

Our hypotheses are that the content of information provided will affect the liking of cultured meat, the willingness of the consumers to accept it and how consumers value the product.

## Material and methods

### Study design

Details of the study design, including the questionnaires, are provided in S1 Text. In brief, a group of 193 participants from the Limburg region of the Netherlands were recruited through a pre-existing online survey cohort. They were informed that they would participate in a meat study and tasting. Participants were representative of the Dutch population in terms of age and sex. Their willingness and eligibility to participate were assessed with an inclusion questionnaire. Exclusion criteria were inability to chew, medication or a condition affecting taste, smell or ability to concentrate or inability to come to the University. Vegetarians and vegans were also excluded. From a total of 903 respondents, 383 were interested and matched the inclusion criteria (i.e. absence of exclusion criteria). From these, based on power considerations, a final sample of n = 193 participants was selected, 41.2% male and 58.8% female, average age = 56 (range 24–84). The sample size of the study was calculated based on 1-way ANOVA, with 1-β = 0.8, α = 0.05, number of between group comparisons = 3, and an effect

size of 0.75. This required n = 59 per group. The study was performed from June-August 2017. The study was approved by the Ethical Review Committee Psychology and Neuroscience (ERCPN, dossier 180_02_06_2017) of the Maastricht University, and all participants were provided with a written informed consent. Participants were debriefed after completion of the total study and were paid for participation.

## Material and preparation

Hamburgers used for this study were frozen beef hamburgers (>99% beef), kindly provided by Zandbergen World's Finest Meats (Zoeterwoude, Netherlands) in June 2017. They were stored at -18˚C and thawed at 2˚C the day before each sensory evaluation. Hamburgers were fried in a pan on an induction heat source with monitored temperature (Hendi 3500D, Steenoven, Netherlands). The pan was preheated for 10 min and the temperature was maintained at 165˚C. The hamburgers were cooked 8 min to reach 73˚C. After cooking, the hamburgers were cut and immediately served to the participants on warm plates. One sample was presented as coming from a conventional hamburger (henceforth termed conventional hamburger), the other from a cultured meat hamburger (termed 'cultured' hamburger), although both samples were conventional meat of equal quality and identically cooked. The hamburgers were cut in two different shapes: a third of a hamburger for the conventional hamburger samples and a quarter for the 'cultured' hamburger samples. The plates were labelled 'conventional hamburger' for the conventional meat sample and 'cultured hamburger' for the 'cultured' meat hamburger.

The study was performed in a room of the Metabolic Research Unit Maastricht (MRUM) at the Maastricht University (room temperature: 22˚C) under white light.

## Product information manipulation and protocol

To avoid priming, the participants filled out an initial questionnaire about their concerns on environmental issues, animal welfare, food security and safety of food products a few days before coming to the University.

The day of the evaluation, the participants received a brief basic technical description of cultured meat and answered the first questionnaire focusing on acceptance of cultured meat (Fig 1). To measure the effects of product information on meat acceptance, participants were divided into three different information-receiving groups (the 'Information Condition'), matched for age and sex. Group 1 received information on societal benefits (n = 65; age, mean (SD) = 56.9(13.7); male:female ratio = 1.71); group 2 received information regarding personal benefits (n = 64; age = 56.4(15.9); male:female ratio = 1.21) and group 3 was informed about meat quality and taste (n = 64; age = 54.7(14.5); male:female ratio = 1.40).

The information given to each group:

Group 1: Societal Benefits

Meat is tasty, but its production through livestock farming will become a major problem in the near future. The world population is growing and the demand for meat is increasing due to increased prosperity, especially in India and China. The increase in demand cannot be met by animal husbandry. Moreover, we know that animal husbandry is responsible for 15–20% of all greenhouse gas emissions, especially methane from cows. Finally, intensive animal husbandry is accompanied by animal suffering. Cultured meat offers a solution for all those problems. Through more efficient use of raw materials, cultured meat requires 90% less land, 90% less water and 60% less energy than meat from livestock. Because we need fewer cows, greenhouse gas emissions are greatly reduced. Reducing deforestation, giving agricultural land back to nature and reducing the greenhouse effect, can stop climate change and perhaps reverse it.

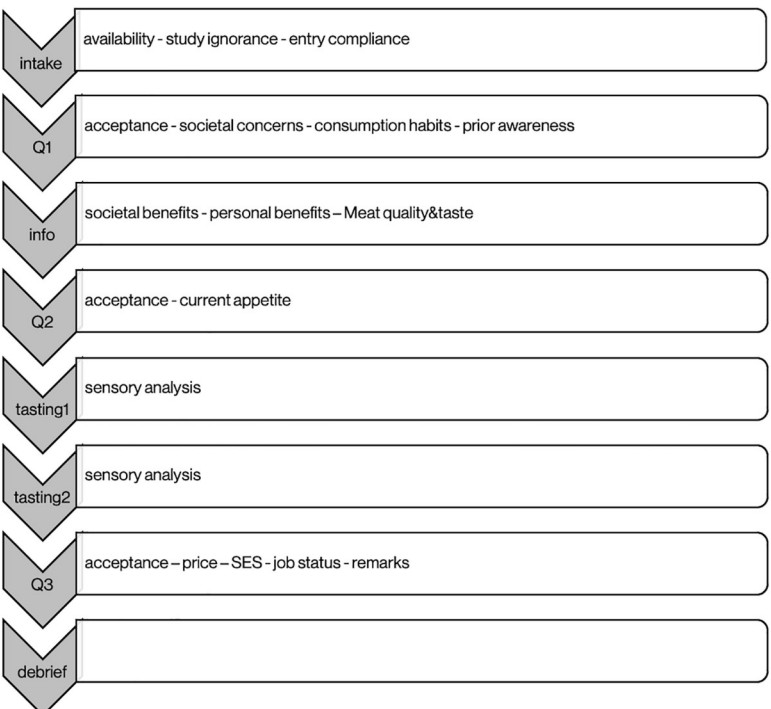

**Fig 1. Study design.** After an intake by telephone, participants came in and answered three questionnaires (Q1, Q2, Q3), before and after having received specific information (info) and after tasting two pieces of hamburger (tasting 1&2). After completion of the study, the participants were debriefed on the nature of the tasted products and the nature of the study (debrief). The in-house part of the study lasted approximately 1 hour.

The smaller volume of livestock radically changes the way animals are kept: small-scale, animal-friendly and without the use of antibiotics or growth hormone. Animals do not have to be slaughtered. You can therefore continue to eat meat without having to worry about these problems.

Group 2: Personal benefits

In terms of composition, cultured meat is the same as regular meat and it is safe; it is made from the muscle stem cells of a healthy, organic cow. That also means that cultured meat has the same nutritional value as regular meat. The controlled culturing method ensures that cultured meat is not contaminated with bacteria and has no disease such as mad cow disease. Moreover, cultured meat contains no antibiotics or hormones and is not genetically manipulated. By raising meat, the quality of each piece of meat can be guaranteed and standardized. The fat content and type of fat—such as polyunsaturated fatty acids or omega-3 fatty acids— can be determined and composed as desired. This would make the meat even healthier than the current product. Cultured meat is safe and approved by the national food safety authority.

Group 3: Meat quality & taste

Eating meat is an experience that is seen as natural and necessary. Cultured meat is the only alternative to regular meat that consists of real meat. It therefore has the same taste, odor, tenderness, juiciness and mouthfeel as regular meat. You can determine for yourself how much fat you want in your meat and how it is distributed (marbled). The quality can be guaranteed because the production is standardized. Meat specialists select the best and most tasty ingredients to make your favorite meat. Cultured meat can meet the requirements of all types of consumers: from the gastronome who wants high quality and who loves adventurous meats to the

fast-food consumer who wants easy to prepare or ready-to-consume meat. You can enjoy every piece of meat: your favorite burgers, steak or prime rib, BBQ meat with your family during holidays. We can offer meat from all types of cow breeds, each with their own specific aroma. Cultured meat contains all the food components of normal meat and is an excellent source of protein.

After providing the information, the acceptance questionnaire was repeated. Upon completion of the second questionnaire, all participants conducted a sensory evaluation of the conventional and 'cultured' hamburger samples and filled out the acceptance questionnaire for a third time. Then, participants were asked to value the cultured hamburger by willingness to pay a premium price. In addition, questions about socio-economic status were asked. Finally, participants were invited to make free-format remarks on cultured meat. The total study session lasted 1 hr.

## Sensory perception analysis

To address the question if the provision of specific product (meat) information influences sensory perception of 'cultured' meat, a wanting/liking test was done on two samples of hamburger, one labeled (on the plate) as 'conventional' and the other one, of different size, as 'cultured'. Samples were offered in random order.

## Data analysis

**Variables.**    Acceptance, liking and circumstantial variables were acquired and evaluated.

Acceptance was defined as the response to four different acceptance questions: 1. "Are you in favor of cultured meat?" ('favor' question), 2. "Do you want to taste it?" ('taste' question), 3. "Will you buy it?" ('buy' question) and 4. "Do you want to replace conventional meat with cultured meat in your daily diet?" ('replace' question). The questions were asked before and after specific information was given, and after tasting in a repeated measures design. Responses to the questions were given on a 5-point scale ranging from 1–5; definitely against to definitely in favor (for the 'favor' question) or definitely not to definitely yes for the 'taste', 'buy' and 'replace' questions. Based on the results of a Reliability test and Cronbach's Alpha scores, the responses to the 'favor', 'buy' and 'replace' questions were summated into a general 'Acceptance' variable, henceforth named *acceptance*. The scores for *acceptance* consequently ranged from 3 to 15. The 'taste' responses were left out as Cronbach's Alpha was not reduced by leaving it out and because the 'taste' question was not asked after the participants had tasted the pieces of hamburger. The response to the 'taste' question ranged from 1 to 5 and was designated as *willingness-to-taste* variable. Cronbach's Alpha was 0.892 for *acceptance* before information was given, 0.901 after information and 0.930 after tasting.

Two additional variables were measured to assess the value participants place on cultured meat: their *willingness to pay a premium price* (yes/no) and *the level of premium price*, they were willing to pay.

Sensory perception variables included *appearance*, *smell*, *color*, *taste*, *tenderness* and *juiciness* on a 7- point hedonic scale. The *presence of an aftertaste* and the *quality of the aftertaste*, were noted as binary variables. All liking variables were acquired for conventional and 'cultured' meat in a paired setup, randomized for order in which samples were tasted.

Circumstantial variables were: *Sex* (male, female), *age*, *family income* (4 categories from <€1000/mo to >€4000/mo), *general education* (5 categories from vocational to PhD level), *prior awareness* about cultured meat (3 categories: 'not aware', 'aware but don't know exactly what is', 'aware and know exactly what it is'), *concerns about animal welfare*, *environment*, *food safety* and *food security* (6 categories from 'not concerned at all' to 'extremely concerned'),

*meat* or *beef eating behaviour* (8 categories from 'less than once a month' to '2 or more times per day') and *appetite before tasting* (wanting savory snack 'yes/no', hungry 'yes/no').

## Analysis

All data were first analysed on accuracy of data entry and on missing values. In case of parametric testing, normal distribution was evaluated using the Shapiro-Wilk test of normality.

*Acceptance* (summation of responses to 3 acceptance questions) was analysed by repeated measures analysis of variance (ANOVA) with *Information Condition* (Societal vs Personal vs Meat Quality & taste) as between factor and *Time* (before and after receiving information and after tasting) as within-subjects measure. The same analysis was performed for the relationship between *acceptance* and *prior awareness* with *prior awareness* as between factor and *time* as within-subjects factor. For Sensory Perception, repeated measures ANOVA included *Information Condition* as between factor and *Type of Meat* as within-subjects measure.

To analyse relationships between levels of *acceptance* before information was provided (entire cohort, n = 193) and circumstantial variables, one-way ANOVA was used with posthoc testing according to Tukey. For the relationship between *age* and *acceptance* (entire cohort, n = 193), Spearman's correlation was used.

To analyse which factors, alone or in combination, predicted *acceptance* and the *level of premium price* participants were willing to pay, stepwise linear regression was performed with responses to *acceptance* (after specific information and before tasting) as outcome measure and *age*, *sex*, *prior awareness*, *Information Condition* and *highest level of education* as predicting factors. For *level of premium* participants are willing to pay, *family income* was added as independent factor. For binary outcome parameters such as presence and quality of aftertaste, the $\chi^2$ test was used to test the effect of *Information Condition* and the McNemar test (paired) for the difference between the *Type of Meat* Condition.

Where appropriate, we presented data as mean and standard deviation: mean(SD).

A P-value of $< 0.05$ (two-sided) was used to define statistical significance.

## Results

### Cohort characteristics

The demographic characteristics of the cohort were similar to that the same age categories in the general population of the Netherlands (Table 1). The average age was somewhat higher in the cohort than the general population and men were slightly overrepresented. Professional and educational status were similar.

### Effect of information content on acceptance

One of the two main questions of the study was if the content of information affects the acceptance of cultured meat. A first Repeated Measures Analysis with *Information Condition* as between-subjects factor and *Time* (before, after information and after tasting) as within-subjects factor on *acceptance* revealed an increase in *acceptance* with *Time* (before and after information: df = 1, F = 109.12, p<0.001; after information and after tasting: df = 1, F = 102.35, p<0.001, Table 2). There was no effect of *Information Condition* (df = 2, F = 0.58, p = 0.561). However, a significant interaction was observed between *Information Condition* and *Time* between before and after information (df = 2, F = 4.13, p = 0.018), but not between after information and after tasting (df = 2, F = 0.27, p = 0.767). This suggests that the *change in acceptance* as a result of the received information, differs between the groups. Indeed, we found the highest mean *change in acceptance* of 1.57(1.77), p = 0.018 in the 'Personal Benefits' group,

**Table 1. Demographic and social economic status of cohort compared to Netherlands.**

| | | cohort | NL[a] |
|---|---|---|---|
| Age | | 56.0±14.7 | 49.0±0.3 |
| Gender | Male | 58.5% | 49.7% |
| | Female | 41.5% | 50.3% |
| Highest level of education? | Unknown | 0% | 2% |
| | Lower vocational education | 2% | 9% |
| | Secondary vocational education | 27% | 19% |
| | Higher professional education | 44% | 38% |
| | University education | 23% | 21% |
| | Post graduate education | 4% | 12% |
| Current professional status? | Wage employed | 42% | 57% |
| | Self-employed | 7% | 14% |
| | Out of work and looking for work | 3% | 2% |
| | Out of work but not currently looking for work | 4% | < 1% |
| | Homemaker | 3% | < 1% |
| | Student | 5% | 3% |
| | Retired | 30% | 21% |
| | Unable to work | 7% | 2% |
| Working in meat industry? | Yes | 1% | |
| | No | 99% | |

a) NL: data for Netherlands were retrieved from the Central Bureau for Statistics Netherlands for 2019 [11]

followed by a non-significant difference in the 'Societal benefits' group (0.97(1.47), p = 0.063) and a smaller *change in acceptance* in the 'Meat quality&taste' group with 0.86(1.23), which was significantly different from the other groups, p = 0.022 (Fig 2).

   The responses to the question "Are you willing to taste cultured meat", or *willingness-to-taste*, also showed significance for the *Time Condition* (df = 1, F = 18.88, p<0.001), but no effect of the *Information Condition* (df = 2, F = 0.09, p = 0.918) and no interaction (df = 2, F = 2.45, p = 0.089).

## Effect of prior awareness on acceptance

From preliminary analysis and from previous studies [12] it is clear that prior awareness is related to acceptance of cultured meat.

**Table 2. Acceptance of cultured meat and Information provision.**

| | Information Condition | | |
|---|---|---|---|
| *acceptance* | **Societal benefits** | **Personal benefits** | **Meat quality & taste** |
| before info | 10.20(2.44) | 10.14(2.72) | 10.69(1.96) |
| after info | 11.17(2.32)* | 11.75(2.33)*# | 11.55(2.01)* |
| after taste | 12.00(2.35)¶ | 11.98(2.80)¶ | 12.36(2.37)¶ |

Values represent mean(SD) scores on the composite variable *acceptance* (range 3–15, see methods). Higher values indicate better *acceptance*.

*: Significant Time effect, before and after information provision

¶: Significant Time effect, before and after tasting

#: Significant Interaction between Information Condition and Time for Personal Benefit group

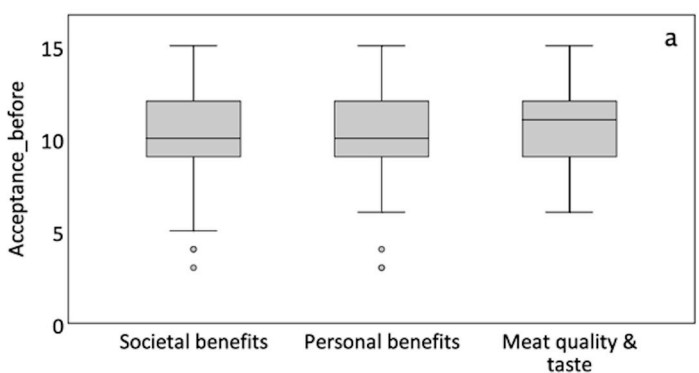 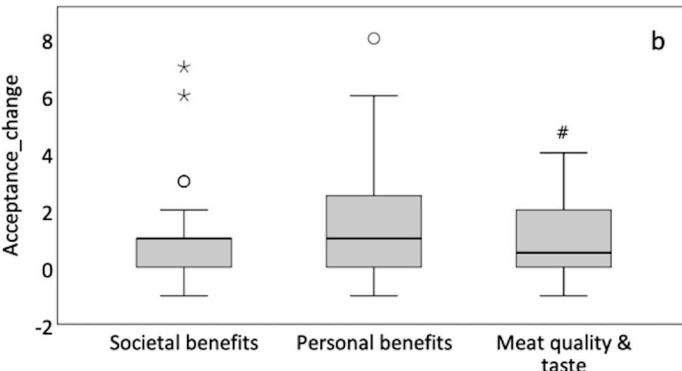

**Fig 2. Boxplot change in acceptance of cultured meat after specific information.** The dark line in the box indicates median value. Open circles: outliers, asterisk: extreme value. #: Significant change in acceptance, p = 0.022.

A second Repeated Measures Analysis with *Awareness Condition* as between-subjects factor and *Time* as within-subjects factor on *acceptance* showed a clear effect of *prior awareness* (df = 2, F = 96.353, p<0.001 for *Time* and df = 2, F = 15.69, p<0.001 for *prior awareness*). Based on the interaction result, the difference in *acceptance* over time (as a result of information provision), was significant for the participants who never heard of cultured meat or did not know exactly what it was, but not for those who already knew exactly what it was. Knowing exactly what cultured meat is, consistently led to higher *acceptance* scores than just having heard of it but not knowing exactly what is or never having heard of it (Table 3).

Given the relationship between *prior awareness* and *acceptance*, it is important to note that the level of *prior awareness* in the three information groups was not different (ANOVA, p = 0.307).

*Willingness-to-taste* was also determined by the *Time Condition* (df = 1, F = 28.59, p<0.001) and *Awareness Condition* (df = 2, F = 11.622, p<0.001), and showed a significant interaction (df = 2, F = 5.57, p = 0.004). The increase in *willingness-to-taste* by providing specific information, was 0.57(0.99) in the participants who had heard and knew exactly what cultured meat was, 1.74(1.62) in for those who heard of it, but were not exactly sure what it was (p = 0.002 compared to the first group, p = 0.203 to group who had never heard of cultured meat) and 2.10(2.21) for participants who had never heard of cultured meat.

**Table 3. Acceptance of cultured meat and prior awareness.**

| | Awareness | | |
|---|---|---|---|
| *acceptance* | No | Yes, but don't know exactly what it is | Yes, and know exactly what it is |
| before info | 8.10(2.47) | 9.45(2.08) | 11.34(2.04)§§ |
| after info | 10.19(2.60)* | 11.22(2.15)* | 11.91(2.08)* # |
| after taste | 10.57(3.78)¶ | 11.64(2.45)¶ | 12.72(2.02)¶ |

Values represent mean(SD) on the composite variable *acceptance* (range 3–15, see methods). Higher values indicate better acceptance.

*: Significant Time effect, before and after information provision

¶: Significant Time effect, before and after tasting

§: Between effect (awareness), posthoc: significantly different from "No" (p<0.001) and "Yes, but don't know what it is" (P = 0.001).

#: Significant Interaction between Time (before and after info) and awareness.

## Acceptance, before providing information

To analyse the relationships between *acceptance* at the start of the study, before information was provided, and circumstantial variables such as *sex*, *education*, *family income*, *meat* or *beef consumption behavior* and levels of concern about *environment*, and *animal welfare*, ANOVA was performed. Of these circumstantial variables, only the level of *education* (df = 4, F = 4.83, p = 0.001) and frequency of *meat consumption* (df = 4, F = 5.44, p<0.001) were significantly related to *acceptance*. No relationship was observed with *acceptance* based on the 'willingness to taste' question, since the *willingness-to-taste* was generally very high and therefore not discriminative.

Posthoc analyses (Tukey) of the relationships between *acceptance* and *education* revealed that university educated participants were more accepting of cultured meat than participants who received a professional or secondary vocational education.

The relationship between *meat consumption* (all meats) and *acceptance* showed lower *acceptance* in the group that ate meat once a week than in groups that ate meat 2-4/week (p = 0.009) or more frequently (p = 0.033).

To study the relationship between *acceptance* and *age* we used Spearman's correlation avoiding the somewhat arbitrary binning of age groups. We found a negative correlation between *age* and *acceptance* (Rho = -0.204; p = 0.004), suggesting that acceptance is less at higher age. No relationship between *age* and *willingness-to-taste* was observed (Rho = -0.090; p = 0.213).

## Regression analysis on acceptance

To further test whether *acceptance* changes as a function of *Information condition* predictor, a regression model was designed to account for possible differences in *change in acceptance* as dependent factor between categories of *prior awareness*, *highest level of education*, *age* and *sex*. Stepwise regression analysis revealed that only *prior awareness* was significantly predictive: $F_{(1,189)} = 12.43$; p = 0.001) with $R^2 = 0.062$ and Beta = 0.808. Thus, Regression analysis confirmed that variables other than *prior awareness* had little or no effect on *acceptance* after specific information was received, including the *Information Condition* itself.

The same analysis was performed on the effect of tasting on *acceptance*. Also here, *prior awareness* was the only predictor of *acceptance* ($F_{(1,190)} = 17.85$; p<0.001, $R^2 = 0.086$, Beta = 1.077).

## Sensory perception

The second aim of the study was to ascertain if the content of information on cultured meat affects sensory perception of meat, either conventional or cultured. A repeated measures ANOVA was performed with *Information condition* as between factor and *Type of meat* as within-subjects factor on changes in scores on the Wanting-Liking evaluation of the two hamburger products. Analysis did not reveal any effect of *Information Condition* on 'liking' (p-values ranged from 0.083 for smell to 0.835 for appearance). The analysis showed a main effect for *Type of meat* on taste with higher Liking scores for the 'cultured' hamburger compared to 'conventional' hamburger (df = 1, F = 4.26; p = 0.04). No interaction was observed for taste (df = 2, F = 0.58; p = 0.559). For the other five tested attributes (*appearance*, *color*, *smell*, *tenderness* and *juiciness*), no main or interaction effects were observed (p-values ranging from 0.107 to 0.999).

The *presence of an aftertaste* (yes or no) and *quality of the aftertaste* (bad or good), were scored as binary outcomes. Using $\chi^2$- analysis, the *Information condition* did not affect the *presence of an aftertaste* (p = 0.514 for conventional hamburger, p = 0.918 for the 'cultured'

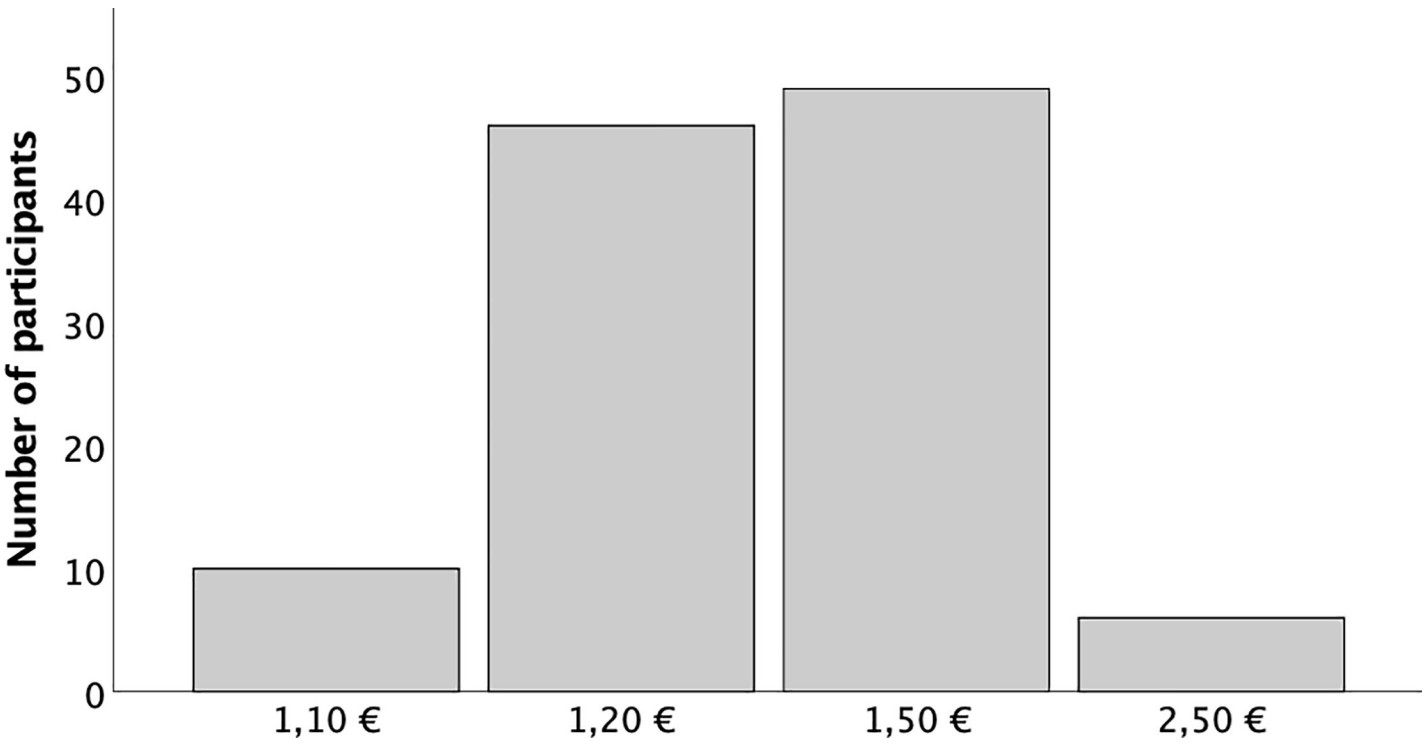

**Fig 3. Willingness to pay a premium price.**

hamburger). The same was found for the *quality of the aftertaste* (p = 0.283 for the conventional hamburger and p = 0.171 for the 'cultured' hamburger).

For the entire cohort, there was no difference in the *presence of an aftertaste* (McNemar test; p = 0.503) or the *quality of the aftertaste* (p = 0.454) between the two hamburgers.

## Premium

As a quantitative measure for the acceptance of cultured meat, we used the willingness to pay a premium price for the product and the increase in amount participants are willing to pay more. A majority (58%) of participants was willing to pay a premium price and this did not depend on the *Information Condition* ($\chi^2$, p = 0.769). On average, the participants willing to pay a premium price were prepared to pay € 0.37 more than the set base price of € 1.00 (Fig 3).

There was also no effect of *Information condition* on the extra amount participants were willing to pay (€ 0.37, € 0.34 and € 0.41 for the 'societal benefits', 'personal benefits' and 'meat quality & taste' groups respectively, df = 2, F = 0.45; p = 0.636). There was no relationship between *family income* and *the level of premium price*, participants were willing to pay (df = 3, F = 0.22; p = 0.881 for entire cohort, df = 3, F = 0.88; p = 0.456, for participants who were willing to pay a premium).

Regression analysis on *the level of premium price* included the independent variables *Information condition*, *prior awareness*, *age*, *education*, *family household income*, and *acceptance* after tasting. None of these factors predicted the *the level of premium price*.

## Solicited comments on cultured meat

On the open-ended question 'What kinds of thoughts do you have about cultured meat? (This included unfinished thoughts, such as doubts, worries, hopes or mixed feelings)' a variety of

positive comments, concerns and neutral comments were expressed by the participants (Fig 4). These comments were categorized by two independent observers. First, the general tone of the comments was qualified as positive, neutral or negative. Then, specific comments or questions were categorized according to their topic (price, health, animal welfare, safety, taste, technical aspects, specific societal aspects) and within that topic qualified as positive, neutral or negative. Inter-rater variability for the general categories positive, neutral and negative was good with kappa values of 0.93, 0.83 and 0.83, respectively, P<0.001). For the four to five most prevalent remarks in each general category, kappa values ranged from 0.67 (positive comment on safety) to 0.903 (negative comment on price), with an average kappa of 0.82(0.07) (mean (SD)).

Positive remarks highlighted mostly the perceived benefits for environment, food security and animal welfare (Fig 4A). Taste was judged positively but this was obviously a result of the 'cultured' hamburger being equal to the conventional one. Food safety was also mentioned as a positive quality of cultured meat. The most prevalent concerns (Fig 4B) were about the anticipated high price, safety and sense of unfamiliarity. The need for regulation was the 4th most prevalent remark. Neutral remarks (Fig 4C) were expressed as questions and referred to health, job loss for farmers, nutritional value and a desire to get more information about cultured meat. There were no remarkable differences in remarks made by participants assigned to the three specific information groups.

## Discussion

We conducted the first acceptance study on cultured meat that was performed in the setting of a tasting of meat that was labeled as either 'conventional' or 'cultured'. Studies thus far have been either focus group studies or questionnaire-based surveys, without a physical product exposure. The study was performed in a representative cross-section of the population that was geographically located in the Limburg region of the Netherlands. We observed that increasing literacy of participants by giving them information on cultured meat increased acceptance. The content of the information was also relevant. In a repeated measures analysis, a modest but significantly higher level of acceptance was observed when participants received information on personal benefits of cultured meat rather than information on quality and taste of the meat. No difference was observed with information on societal benefits.

Initial acceptance based on the question if people are in favor of the cultured meat development, was already fairly high before specific information was provided, ranging from 48 to 56% in the studied groups. These results are in accordance with earlier quantitative studies that were geographically spread through Europe [3, 13, 14], the US and parts of Asia [4, 15]. Acceptance of novel, biotechnical, foods before any given information depends on prior awareness about the subject [4, 10, 12, 16–18] and indeed we observed a strong positive relationship between prior awareness/understanding and initial acceptance rate of cultured meat. The studied population had a higher level of prior awareness (89% had heard of it; 55% knew what it was) than the population in Belgium (49% heard of it) [3], and the US (29% heard of it; data not reported but calculated from the online available source data) [4]. Bekker et al observed that prior awareness negated the effect of positive and negative information, respectively on positive or negative explicit attitudes towards cultured meat [10]. There can be a complex relationship between prior awareness/information and the impact of newly provided information, where participants basically weigh the two [12]. The strength of prior beliefs and information, especially when thinking about future technologies with no tangible products in the marketplace yet, can be as important as the new information [17]. This was illustrated in a study on the effect of information on the acceptance of genetically modified food, where the

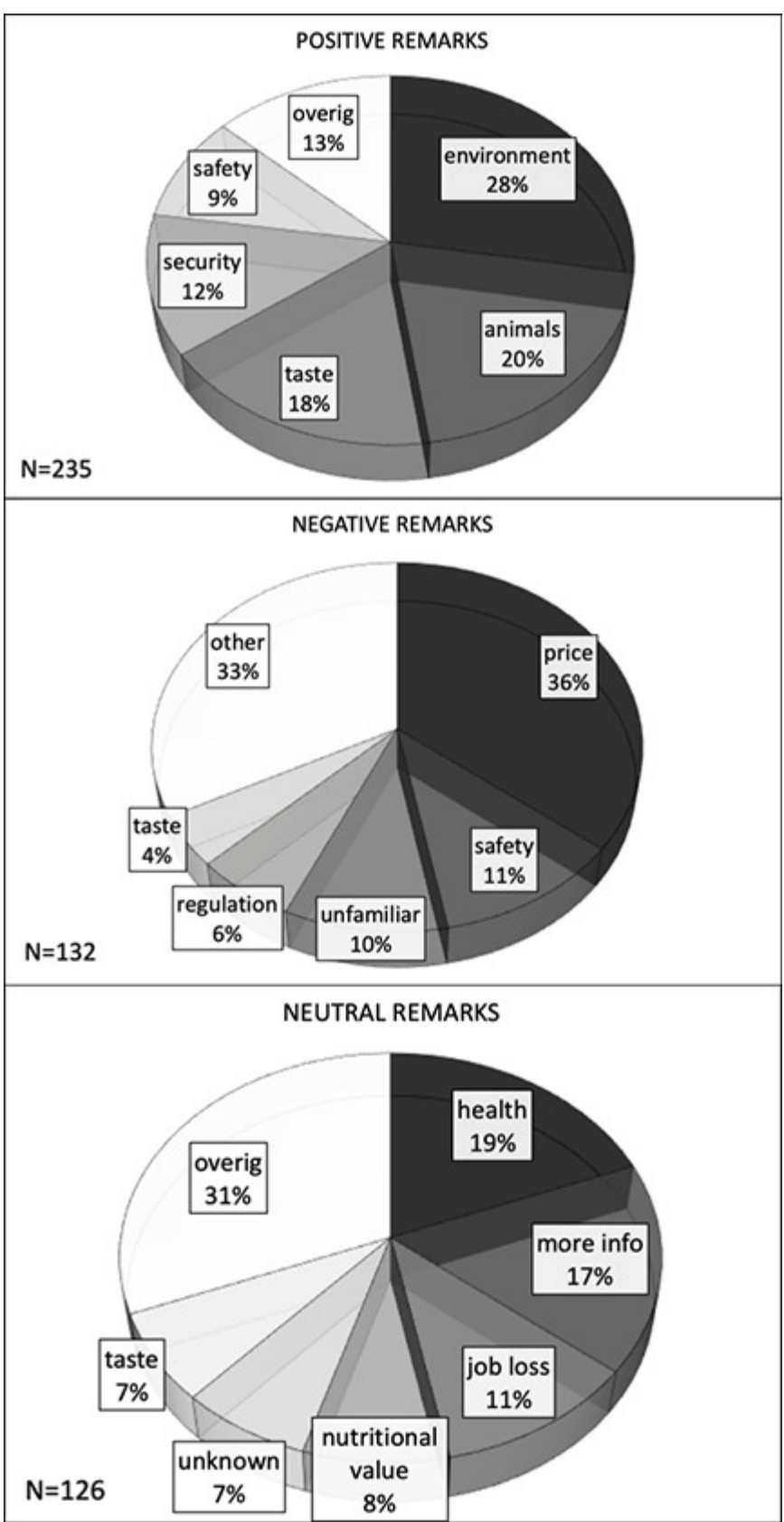

**Fig 4. Qualitative remarks.** Positive (a), negative (b) and neutral (c) remarks, on cultured meat. N = number of remarks per category, the size of the pies indicates the relative number of remarks on the specified topic.

same information had opposite effects in different geographical locations, presumably because some people did not trust the provided information [19]. Independent third-party information has more impact than company-provided information. We considered the university-provided information as independent, although some participants may have viewed the local university as an interested party since it has been publicized as the birthplace of cultured meat. Thus, it is possible that populations less well informed than the participants in our study, may respond stronger to the content of information provided. It is difficult to judge if the participants experienced the information provided to them in an academic setting was considered as third-party information, as that question was not specifically addressed.

We chose to give only positive information on cultured meat. It is expected that more balanced information would affect acceptance differently [20]. For instance, Rousu et al. found that third party verifiable information had more effect when negative pre-existent information had been present on the acceptance of genetically modified food [21].

The steady increase in acceptance after information and after tasting confirms earlier studies on unfamiliar foods [22–24]. This is the first study on acceptance of cultured meat with a tasting component, which allows to study of the effect of physical exposure to an otherwise imagined product on acceptance. In a recent study of Tan et al, where participants were presented with hamburgers containing unfamiliar components such as lamb brain, frog meat or mealworms, the tasting experience did not lead to the hamburgers being accepted as appropriate food [16]. Disqualifying novel foods by considering them inappropriate likely reduces commitment to start adopting them. In our study however, tasting did increase acceptance. This might suggest that a cultured meat hamburger is considered an appropriate food when its sensory features are equivalent to conventional meat.

It is interesting to note that the four acceptance questions produced different results. Specifically, the willingness to taste cultured meat had a much higher score than the responses to the other questions. The same difference was found by Wilks et al [4], where two-third of the participants was willing to try in vitro meat and only one-third willing to eat it on a regular basis. Likewise, in a large Italian cohort, willingness to try (54%) scored higher than willingness to buy (44%) or to pay more (23%) [14]. Willingness to taste was further substantiated by the observation that all participants actually tasted the 'cultured' hamburger that was presented to them. A similar observation was made by Tan et al where 94% of the participants tasted the hamburgers that were labelled to contain unfamiliar items [16] and by Caparros Medigo et al. in whose study 70–90% of participants ate insects, with the level being age-dependent [25]. In the Caparros Medigo study, willingness to taste was situation dependent (during 'special' occasions that elicited curiosity). It is therefore possible that the study context and university environment of our study may have had a positive effect on the decision to taste the 'cultured' hamburger. As perceived danger is a major determinant for willingness to taste novel foods [26], this suggests that participants did not consider cultured meat dangerous. Unwillingness to try novel foods is generally referred to as neophobia. We did not test food neophobia in this particular population, but in previous studies food neophobia appeared to be a poor predictor of choices for unfamiliar foods [16, 27]. However, recent studies by Bryant et al [15] and by Wilks et al [28] have identified food neophobia as a predictor of cultured meat rejection. Disgust and perceived unnaturalness seem to underlie unwillingness to try foods that are considered technical [10]. Indeed, disgust and perception of naturalness appear to play a role in rejection of cultured meat as well [29].

The sensory evaluation of the hamburgers on the six attributes revealed a difference between the conventional and 'cultured' hamburger on taste, which was considered slightly better for the 'cultured' hamburger. On the other attributes, global appearance, color, smell, tenderness, juiciness and aftertaste, there were no differences between the two hamburgers. The content of information provided had no influence on sensory analysis. Several theories and experimental data relate sensory expectations to sensory analysis [12]. Expectations can be created by providing information, but they are typically referring to taste or quality of a product. Since we found no difference in taste evaluation between the information groups, and therefore not a higher taste difference between the conventional and 'cultured' hamburger in the 'Meat quality & taste' information group, our data cannot be directly interpreted along those theories and experimental data. The label 'cultured hamburger' probably did create expectations, more likely towards an inferior taste experience than a superior one [4, 30]. Given that the hamburgers were identical, the taste experience of the 'cultured' hamburger was a pleasant one and thus led to a positive valuation through assimilation [31], whereas the experience and expectation of the conventional hamburger probably matched. In a similar experimental setup, Tan et al found a strong label effect on sensory-linking of hamburgers with unfamiliar components [16]. In that study however, the hamburger with unfamiliar components was generally rated lower on the sensory-liking scale but they also did have a composition that was different from the beef hamburger. For experimental reasons, we presented the conventional hamburger in a slightly larger size than the 'cultured' piece, justified by 'limited availability' of the latter. This may have affected the sensory analysis outcome as items perceived as scarce can be considered of higher value [32].

Interpretation of the sensory analysis outcome depends on the confidence participants had in the authenticity of the 'cultured hamburger'. We have not addressed this in the questionnaire to avoid priming when asked before tasting (raising suspicion) or unreliable answers after tasting (not admitting being misled). Based on reactions during debriefing and the absence of remarks expressing this suspicion in the solicited comments provided sufficient confidence that participants believed to have eaten a piece of cultured hamburger.

A surprising result was that more than half of the participants were willing to pay a premium for cultured meat with an average of 37% more than for conventional meat. We used this parameter as a measure of acceptance. Many studies in food acceptance have used a methodology called experimental auctioning, where groups of participants are bidding on food items either to acquire the food items (willingness to pay) [17] or to accept the foods (willingness to accept; receiving compensation for an undesirable food) [19]. In the study by Wilks et al, without a tasting experience, only 16% was willing to pay a premium [4]. In the Verbeke study, 14% was willing to pay a premium after having received basic technological information, which number rose to 36% when additional information on benefits of cultured meat had been provided [3]. Baseline willingness to pay a premium for cultured meat was 23% and the premium range was between 10 and 30% in an Italian study [14]. We observed no influence of specific information on willingness to pay a premium, in contrast to the study of Verbeke. It is possible that the favorable tasting experience led participants to be more enthusiastic resulting in willingness to pay a premium. Alternatively, the generally more favorable attitude towards cultured meat in comparison with previous studies may be associated with a higher willingness to pay a premium. Since we did not ask the question before and after tasting, this question remains unanswered. It does suggest however, that consumers are potentially willing to pay a premium for cultured meat based on its perceived benefits.

All studies on the acceptance of cultured meat has been survey-based or performed in focus groups. The information from these types of studies can be subject to fallacies, the most important one perhaps being the conscious choice fallacy. Surveys select for conscious choices where

consumer behaviour is not always conscious, leading to a discrepancy between responses in surveys and actual behaviour [33]. In the absence of a commercial product, the approach of exposing participants to a labeled product which they believe to be cultured meat, is likely closer to the human experience. However, this study was still performed in a rather formal academic setting that is different from the regular food purchase setting.

Some of the favorable responses to acceptance and willingness to pay questions were received after the favorable tasting experience, which is obviously artificial, as the tested product was in fact conventional beef. These results can therefore only be translated to an eventual cultured meat product, if that is indistinguishable from a conventional hamburger. That result has to be awaited.

## Conclusion

For a novel and unfamiliar food such as cultured meat, acceptance depends on the level of information, either pre-existent or provided ad hoc. The content of information seems of minor importance. When framed positively and when tasting experiences are favorable, acceptance of cultured meat is potentially high. The perceived benefits of cultured meat may translate in a willingness to pay a premium price.

## Supporting information

**S1 Text. Online invitation questions and questionnaire.**
(DOCX)

## Acknowledgments

The authors are grateful for the advice of prof Catherine Dacremont, ENSBANA, Université de Bourgogne, France and prof Cor van der Weele. They are also indebted to the organizational support by Ms Bianca Gorski, Ms Vivian Schellings, Mr Samuel Becker, Ms Sophia von Stockert and Ms Frea Mehta. Dr Ton van Ambergen provided valuable statistical guidance and dr Josh Flack critically read and edited the manuscript.

## Author Contributions

**Conceptualization:** Nathalie C. M. Rolland, C. Rob Markus, Mark J. Post.

**Data curation:** Nathalie C. M. Rolland, Mark J. Post.

**Formal analysis:** Nathalie C. M. Rolland, C. Rob Markus, Mark J. Post.

**Funding acquisition:** Mark J. Post.

**Investigation:** Nathalie C. M. Rolland.

**Methodology:** Nathalie C. M. Rolland.

**Project administration:** Nathalie C. M. Rolland.

**Supervision:** C. Rob Markus, Mark J. Post.

**Validation:** C. Rob Markus.

**Writing – original draft:** Nathalie C. M. Rolland, C. Rob Markus, Mark J. Post.

**Writing – review & editing:** Nathalie C. M. Rolland, C. Rob Markus, Mark J. Post.

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
