## [Decision Letter · Decision Letter 0]

7 Jan 2020

PONE-D-19-34685

The effect of information content on acceptance of cultured meat in a tasting context

PLOS ONE

Dear Head of Physiology Post,

Thank you for submitting your manuscript to PLOS ONE. After careful consideration, we feel that it has merit but does not fully meet PLOS ONE’s publication criteria as it currently stands. Therefore, we invite you to submit a revised version of the manuscript that addresses the points raised during the review process.

I really appreciate for your submitting an interest paper to PlosOne.

Both reviewers confirmed importance and impacts on your manuscript, but both of them finds some problems to be fixed before published. All points can be addressed in next manuscript, I hope.

Also I strongly recommend you to be checked and corrected your manuscript by a professional English native speaker.

We would appreciate receiving your revised manuscript by Feb 21 2020 11:59PM. To enhance the reproducibility of your results, we recommend that if applicable you deposit your laboratory protocols in protocols.io, where a protocol can be assigned its own identifier (DOI) such that it can be cited independently in the future. For instructions see: http://journals.plos.org/plosone/s/submission-guidelines#loc-laboratory-protocols

We look forward to receiving your revised manuscript.

Kind regards,

Nobuyuki Sakai, Ph.D.

Academic Editor

PLOS ONE

Journal Requirements:

3. Please ensure that you refer to Figure 2 in your text as, if accepted, production will need this reference to link the reader to the figure.

'The research in this paper was made possible by a gift from the Mark Post Research

Foundation, The Netherlands.'

'Yes, MJP received an award from Stichting Wetenschappelijk Onderzoek Limburg. It has no grant number. URL: https://www.ufl-swol.nl. The sponsor did not play a role in the study'

Please provide an amended Funding Statement that declares *all* the funding or sources of support received during this specific study (whether external or internal to your organization) as detailed online in our guide for authors at http://journals.plos.org/plosone/s/submit-nowPlease state what role the funders took in the study.  If any authors received a salary from any of your funders, please state which authors and which funder. If the funders had no role, please state: "The funders had no role in study design, data collection and analysis, decision to publish, or preparation of the manuscript."

'I have read the journal's policy and the authors of this manuscript have the following competing interests: Mark Post is Chief Scientific Officer and co-founder of Mosa Meat, a company that aims to commercialise cultured meat.'

6. Please include captions for your Supporting Information files at the end of your manuscript, and update any in-text citations to match accordingly. Please see our Supporting Information guidelines for more information: http://journals.plos.org/plosone/s/supporting-information

Reviewers' comments:

Reviewer's Responses to Questions

**Comments to the Author**

1. Is the manuscript technically sound, and do the data support the conclusions?

Reviewer #1: Yes

Reviewer #2: Yes

2. Has the statistical analysis been performed appropriately and rigorously? 

Reviewer #1: Yes

Reviewer #2: Yes

3. Have the authors made all data underlying the findings in their manuscript fully available?

Reviewer #1: Yes

Reviewer #2: Yes

4. Is the manuscript presented in an intelligible fashion and written in standard English?

Reviewer #1: Yes

Reviewer #2: Yes

5. Review Comments to the Author

Reviewer #1: This is a very important study which addresses the major drawback of cultured meat acceptance research thus far, namely its hypothetical nature. By having participants actually eat (what they believe to be) cultured meat, this is overcome. The study does a fantastic job of adding to the cultured meat consumer acceptance literature, and I have only minor comments.

P. 11 Effect of prior awareness on acceptance - reference [17] would show this more clearly than [11]. Next sentence, you have a crucial missing “not” - ‘was significant for subjects who had never heard of cultured meat or did not know exactly what it was, but NOT for those already knew…’

P. 14 Solicited comments on cultured meat - the sentence here in parentheses should be complete - there is also a superfluous apostrophe here.

Reviewer #2: The paper reports a single experiment investigating the effect of information content on consumer’s acceptance of cultured meat. I think this topic may be of interest for the readers of PLOS ONE, but I also think there are some major problems which must be addressed before it is possible to consider it for publication.

1. The authors discussed that the previous studies about culutured meat were conducted with students in agriculture and food sciences, and therefore the present study enployed a general population. In the Participants section, however, there is no information about participant’s faculty or profession. The authors should show information about participant’s faculty or profession in detail and the most of them do not major academic fields related to food sciences.

2. In the experiment, the participants were provided the two samples of the same conventional meat hamburger with different labels, named “cultured” and “conventional” hamburgers. However, it seems that the authors did not check whether their experimental manipulation succeed, i.e. whether the participants did not aware that the two pieces of hamburgers were idential and not used cultured meat. If some participants had awared these deception procedures, the relaiability of the present results will become diminished. The authors should mention this problem.

Minor issues.

3. I think the contents of each information condition may be dscribed not in the Appendix but in the main text of the paper because these are very important infomation in this study.

4. Figure 4. Please add the frequency or ratio (%) of each remark.

5. The words “participants” and “subjects” are mixed in the manuscript. Please use the same expression throughout a manuscript.

6. PLOS authors have the option to publish the peer review history of their article (what does this mean?). If published, this will include your full peer review and any attached files.

Reviewer #1: No

Reviewer #2: No

---

## [Author Response · Author response to Decision Letter 0]

9 Mar 2020

We appreciate the careful reading and thoughtful comments by the reviewers and made the necessary changes to the manuscript. Responses (R) point by point:

Editor

1. The manuscript was read and edited by a native Brit.

2. Please ensure that your manuscript meets PLOS ONE’s style requirements

R: We have made all the changes necessary to conform with the style requirements

3. Phrase “data not shown”, is not permitted.

R: Since the data are provided in detail through a repository (Mendeley), the data are actually retrievable, so we omitted the phrases “data not shown”.

4. Please ensure that you refer to Figure 2.

R: Apologies for this oversight. We now refer to figure 2. 

5. Please remove any funding related text in the manuscript.

R: We removed the reference to funding. In the online submission, we have added the phrase on the role of funders in the study. These statements are also included in the cover letter

6. Competing interest section of online submission.

R: The statement that the Competing Interest has not altered adherence to the PLOS ONE policies on sharing data and materials has been included in the online submission and the cover letter.

7. Please include captions with supporting information.

R: We have added the supporting information to the Supplemental text at the bottom of the manuscript and provided a caption

Reviewer 1

P11. Effect of prior awareness on acceptance - reference [17] would show this more clearly than [11]. Next sentence, you have a crucial missing “not” - ‘was significant for subjects who had never heard of cultured meat or did not know exactly what it was, but NOT for those already knew…

R: The reference has been changed to the one suggested. Thanks for noting the missing ‘not’. We have added it.

P14. Solicited comments on cultured meat - the sentence here in parentheses should be complete - there is also a superfluous apostrophe here.

R: We have completed the sentence within parentheses. The quotation mark at the end of the parentheses is in our view correct as the entire phrase, including the part between parentheses, is part of the quote.

Reviewer 2

1. The authors discussed that the previous studies about culutured meat were conducted with students in agriculture and food sciences, and therefore the present study enployed a general population. In the Participants section, however, there is no information about participant’s faculty or profession. The authors should show information about participant’s faculty or profession in detail and the most of them do not major academic fields related to food sciences.

R:

2. In the experiment, the participants were provided the two samples of the same conventional meat hamburger with different labels, named “cultured” and “conventional” hamburgers. However, it seems that the authors did not check whether their experimental manipulation succeed, i.e. whether the participants did not aware that the two pieces of hamburgers were idential and not used cultured meat. If some participants had awared these deception procedures, the relaiability of the present results will become diminished. The authors should mention this problem.

R: This is a good point. We were aware at the beginning of the study that this could be a problem. In our pilot we made the two hamburger pieces exactly the same in size and shape and got the impression that participants did not believe them to be different. For that reason, we changed the size of the ‘cultured’ hamburger piece, which was not ideal for sensory analysis as this might have emphasized the exclusivity of the product. However, by changing the size, we have the strong impression that participants believed the nature of the ‘cultured’ hamburger, judged by their reactions during debriefing (one participant was so angry that he professed never to participate in a study at our University again). The issue of believing the ‘cultured’ nature of the sample was not formalized in a specific question, because we judged it ambiguous from an experimental point of view. Asking that question before debriefing, might have raised suspicion. After briefing, the question likely had resulted in difficult to interpret answers (e.g. not willing to admit that they were fooled).

We have added a paragraph in the discussion to mention this potential caveat.

Minor issues.

3. I think the contents of each information condition may be described not in the Appendix but in the main text of the paper because these are very important information in this study.

R: we have added the information boxes to the Methods section. This has inevitable increased the manuscript size with 512 words.

4. Figure 4. Please add the frequency or ratio (%) of each remark.

R: We have added the ratio of specific remarks per category to the figure 4.

5. The words “participants” and “subjects” are mixed in the manuscript. Please use the same expression throughout a manuscript.

R: we have replaced subjects by participants, except for the description of repeated measure analysis in within-subject and between-subject analyses. These descriptions were deemed to be standard statistical language.

---

## [Decision Letter · Decision Letter 1]

18 Mar 2020

The effect of information content on acceptance of cultured meat in a tasting context

PONE-D-19-34685R1

Dear Dr. Post,

We are pleased to inform you that your manuscript has been judged scientifically suitable for publication and will be formally accepted for publication once it complies with all outstanding technical requirements.

As your information, the comment from Reviewer #2 would be a good hint for your continuing studies;

According to the response to Comment #1 and #2, I think the authors should show these information and discussions not only in the response letter but also in the manuscript for the readers of PLOS ONE. Why don’t you show these information related to the validity of your study design for your potential readers?

Although I am agree with this comment, I have made a decision of "Accept" because your manuscript is good enough to be appear in PLOS ONE. I am still thinking your study would be higher level, if you add some discussion and/or description answering to the comment in your manuscript.

With kind regards,

Nobuyuki Sakai, Ph.D.

Academic Editor

PLOS ONE

Additional Editor Comments (optional):

Reviewers' comments:

Reviewer's Responses to Questions

**Comments to the Author**

1. If the authors have adequately addressed your comments raised in a previous round of review and you feel that this manuscript is now acceptable for publication, you may indicate that here to bypass the “Comments to the Author” section, enter your conflict of interest statement in the “Confidential to Editor” section, and submit your "Accept" recommendation.

Reviewer #1: All comments have been addressed

Reviewer #2: (No Response)

2. Is the manuscript technically sound, and do the data support the conclusions?

Reviewer #1: Yes

Reviewer #2: Partly

3. Has the statistical analysis been performed appropriately and rigorously? 

Reviewer #1: Yes

Reviewer #2: Yes

4. Have the authors made all data underlying the findings in their manuscript fully available?

Reviewer #1: Yes

Reviewer #2: Yes

5. Is the manuscript presented in an intelligible fashion and written in standard English?

Reviewer #1: Yes

Reviewer #2: Yes

6. Review Comments to the Author

Reviewer #1: The authors have addressed all concerns. The paper will be a very valuable contribution to the literature on cultured meat acceptance.

Reviewer #2: According to the response to Comment #1 and #2, I think the authors should show these information and discussions not only in the response letter but also in the munscript for the readers of PLOS ONE. Why don’t you show these inforamtion related to the validity of your study design for your potential readers?

7. PLOS authors have the option to publish the peer review history of their article (what does this mean?). If published, this will include your full peer review and any attached files.

Reviewer #1: Yes: Christopher Bryant

Reviewer #2: No

---

## [Editor Report · Acceptance letter]

30 Mar 2020

PONE-D-19-34685R1 

The effect of information content on acceptance of cultured meat in a tasting context 

Dear Dr. Post:

I am pleased to inform you that your manuscript has been deemed suitable for publication in PLOS ONE. Congratulations! Your manuscript is now with our production department. 

With kind regards,

on behalf of

Dr. Nobuyuki Sakai 

Academic Editor

PLOS ONE